# Simultaneous Optimization of Phenolic Compounds and Antioxidant Abilities of Moroccan *Pimpinella anisum* Extracts Using Mixture Design Methodology

Meriem Soussi [1], Mouhcine Fadil [2,*], Wissal Al Yaagoubi [1], Meryem Benjelloun [1] and Lahsen El Ghadraoui [1]

[1] Laboratory of Functional Ecology and Environmental Engineering, Faculty of Sciences and Technology, Sidi Mohamed Ben Abdellah University, Fez 30000, Morocco
[2] Physico-Chemical Laboratory of Inorganic and Organic Materials (LPCMIO), Materials Science Center, Ecole Normale Supérieure, Mohammed V University in Rabat, Rabat 10100, Morocco
* Correspondence: m-fadil@um5s.net.ma

**Abstract:** *Pimpinella anisum* (anise) is a dense vegetal matrix with considerable amounts of bioactive components known for its pharmacological properties. The optimization of extraction constitutes an important key to improving efficacy and avoiding wasting time. Within this framework, the present study was designed to select the most appropriate extractor solvent mixture to extract phenolic and flavonoids using Mixture Design Methodology. The concerned responses were the total phenolic content (TPC), total flavonoid content (TFC) and antioxidant ability examined by 2,2-diphenyl-l-picrylhydrazyl (DPPH) assay. Before mixture design optimization, a screening of solvents was conducted on ten polar and nonpolar solvents to choose the best solvents that give a maximum of total phenolic compounds. This first step has shown that water, ethanol and methanol were the best-used solvents. Later, an augmented centroid design investigated the solvent system's optimization. The results of simultaneous optimization have shown that the ternary mixture containing 44% of water, 22% of ethanol and 34% of methanol was the most appropriate for simultaneous maximization of TPC, TFC and antioxidant activity with 18.55 mg GAE/g, 7.16 mg QE/g and 0.56 mg/mL, respectively. Our results have shown that using mixture design as an optimization technique was an excellent way to choose the most suitable mixture to extract bioactive compounds, which may represent a promising method of multi-purpose extraction, especially in the pharmaceutical and food sectors.

**Keywords:** *Pimpinella anisum*; optimization; total phenolic content; total flavonoid content; antioxidant activity

## 1. Introduction

Medicinal plants have occupied a significant role in human food since early history. They are used for many purposes as raw matter in culinary and therapeutic systems, either in traditional or modern medicine [1]. The dense chemical composition of medicinal plants was the main factor for their uses in folk and modern medicine. The therapeutic abilities of medicinal drugs depend on numerous factors, as well as the part of the plant, the nature of the extract, isolated components and purification technique [2–4]. Thus, classical methods of extraction have continually been used in traditional therapy. However, recent studies focus on optimizing the extraction as the most efficacious tool for extracting a high amount of bioactive compounds from plant matrix [5,6]. In this way, several researchers have focused their interest on studying various factors such as extraction method, the effect of extraction time, temperature, the form of a material matrix and the nature of the solvent [7–9]. Actually, there is no magic recipe or mixture of solvents to elevate the yield of extractable bioactive compounds. Therefore, different methods were investigated to find the optimal operating conditions that can lead to the highest number of bioactive compounds. Among these techniques and experimental designs are reliable and precise

chemometric techniques based on the mathematical modeling of a response in terms of the dependent factors [10,11].

Numerous mathematical models have been developed to elevate the amount of extractable bioactive compounds, including response surface methodology (RSM) [12,13] and mixture design [14,15]. These models are an alternative approach for recovering a high amount of bioactive components and allow more than one response to be optimized simultaneously [16]. Optimizing the amount of phenolic compounds seems to be related to several proprieties, such as antioxidant activity. The modeling of the extraction process using response surface methodology is widely used to choose the right amounts of different solvents to recover the maximum yield of pharmacologically active ingredients from a matrix plant [17]. This technique has been used in various fields, including experimental studies in medicinal plants [18]. Indeed, it has been proved that the optimized extract exhibits high antioxidant potency. These findings were confirmed by the chemical profile of the extract, indicating the presence of different pharmacologically active molecules with high antioxidant potential [13,16,19].

*Pimpinella anisum* (*P. anisum*) is a member of *Umbelliferae* family known for its traditional uses in the folk medicine of several civilizations [20]. Mounting evidence has confirmed the ability of *P. anisum* to exert numerous pharmacological properties, including antibacterial [21], antifungal [22], insecticidal [23], anticonvulsant [24], analgesic [24], antidepressant and anxiolytic activities [20,25]. Furthermore, these pharmacological properties are associated with phytochemical content, such as polyphenolic compounds. Indeed, it has been confirmed that *P. anisum* contains numerous phytochemical constituents, including flavonoids and polyphenolic components [26].

Experimental designs are among the most common avenues adopted to optimize phenolic compounds' extraction from different vegetal matrices [27–29]. In addition, other methods have gained increased interest, including microwave and ultrasonic-assisted extraction, thanks to their simplicity, rapidity and low cost [27]. These methods play a pivotal role in examining the possible interaction of different parameters, providing the most appropriate combinations to minimize the time and number of experiments [13,27,30]. Within this framework, the current study is designed to identify the most suitable solvent mixture for polyphenolic compound extraction and antioxidant ability from *P. anisum* using an augmented simplex-centroid design.

The nature of the extractive solvent is considered among the most decisive parameters in phenolics recovery [31]. Indeed, it has been discovered that some solvents' propriety, especially polarity, impacts the extraction of phenolic compounds [32]. In this way, the first step in this work consists of screening ten pure solvents among the most commonly used in the extraction of phenolic compounds to select those with the most suitable extractive properties. The screened solvents include nonpolar ones such as hexane, solvents with low polarity such as dichloromethane, solvents with important polarity such as acetone and acetonitrile and solvents with high polarity such as water, ethanol and methanol. Then, the solvents chosen from this screening were used as constituents of the mixture of the system of solvents. The appropriate mixture was selected based on the optimized high total phenolic-flavonoid content and antioxidant power. To the best of our knowledge, this is the first time that such work has been done on the species *P. anisum*.

## 2. Materials and Methods

### 2.1. Chemicals and Materials

Seeds of anise were collected from El Hajeb region during the autumn season of 2021 (33°41′22″ N, 5°21′13″ W). Ethanol, methanol, dichloromethane, chloroform, acetone, ethyl acetate, hexane, butanol, acetonitrile, water, DPPH, Folin-Ciocalteu reagent, gallic acid and quercetin were the chemicals used in different experiments and were purchased from Sigma-Aldrich (Sigma-Aldrich Co., St. Louis, MO, USA).

### 2.2. Samples Preparation

The plant was identified by the botanist of the National Agency of Medicinal and Aromatic Plants (NAMAP) in Taounate, Morocco. The samples were dried to constant weight in a room where the temperature was fixed at 25 °C and then powdered in a lab grinder machine.

### 2.3. Ultrasonic-Assisted Extraction

Initially, the extraction was performed by ten pure polar and nonpolar solvents, including water, ethanol, methanol, dichloromethane, chloroform, acetone, ethyl acetate, hexane, butanol and acetonitrile. 50 mg of powdered seeds were mixed with 1 mL of each solvent. The mixtures were sonicated for 30 min in a P30 H ultrasonic cleaner bath with a power of 100 W and a frequency of 37 kHz (Elmasonic, Singen, Germany) and centrifuged at 10,000 rpm/10 min in a Hettich ROTOFIX 32A centrifuge (Hettich, Tuttlingen, Germany). The mixture was filtered and stored at 4 °C until experimental analysis, as previously described by Ousaaid et al. [33]. Afterward, an extraction under the same conditions was carried out by a solvent system formulated from the three most effective solvents. The solvent fractions of each experiment are generated from the chosen mixture design.

### 2.4. Total Phenolic Content Quantification

The phenolic content was quantified using the colorimetric methods previously described by Singleton et al. [34]. In brief, 25 µL of extract and 450 µL of Folin-Ciocalteu reagent (0.5N) were mixed. Then, the mixture was treated with 450 µL of sodium carbonate and incubated in the dark for 2 hours. The absorption was read at 760 nm and the results were expressed as mg GAE/g using a gallic calibration curve.

### 2.5. Total Flavonoid Content Quantification

The total flavonoid content was determined by the colorimetric method according to the method described by Laaroussi et al. [35]. The protocol involves mixing 100 µL of extract, 150 µL of Alcl$_3$ reagent (10%) and 200 µL of NaOH (1%). The obtained mixture was incubated in the dark for one hour and the absorbance was read at 510 nm.

The results are expressed as mg of quercetin equivalent per gram (QE/g) of anise seeds anise using a quercetin calibration curve.

### 2.6. Antioxidant Activity

The antioxidant activity of extracts was evaluated using the DPPH test [36]. In brief, the extract and DPPH solution were mixed with the following quantities, 25 µL and 875 µL, respectively. The mixture was vortexed and incubated for one hour in the dark condition. The absorbance of the mixture was measured at 517 nm. The percentage inhibition (PI) of the DPPH radical by the samples was calculated according to the formula:

$$PI(\%) = \left( \frac{control\ absorbance - sample\ absorbance}{control\ absorbance} \right) \times 100 \qquad (1)$$

The antiradical activity was expressed as DPPH$_{IC50}$ (mg/mL), which is the exact concentration dose required of extractable DPPH$_{IC50}$ whose value corresponds to a higher antioxidant activity.

### 2.7. Mixture Design

The current study used a mixture design methodology to optimize the extraction process. The test is based on combining different independent factors to create a set of equations that donate theoretical values [13]. The solvent system components are presented in Table 1. According to this table, each of the three components of the mixture can have a value between 0 and 100% and the sum of the three proportions will always be equal to 100%. Besides, 12 experiments were performed to examine the impact of solvent

proportions on total phenolic and flavonoid content and antioxidant activity, as described in Figure 1.

**Table 1.** Identification of solvent system factors.

| Components | Level − (%) | Level + (%) | Coded Variables | Level − | Level + |
|---|---|---|---|---|---|
| Water | 0 | 100 | X1 | 0 | 1 |
| Ethanol | 0 | 100 | X2 | 0 | 1 |
| Methanol | 0 | 100 | X3 | 0 | 1 |
| Sum of proportions | 100 (%) | | | 1 | |

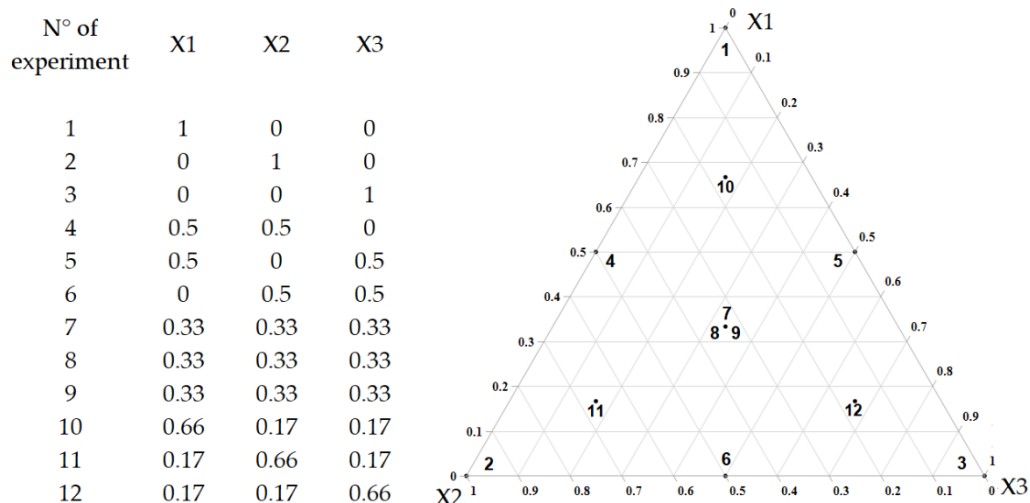

| N° of experiment | X1 | X2 | X3 |
|---|---|---|---|
| 1 | 1 | 0 | 0 |
| 2 | 0 | 1 | 0 |
| 3 | 0 | 0 | 1 |
| 4 | 0.5 | 0.5 | 0 |
| 5 | 0.5 | 0 | 0.5 |
| 6 | 0 | 0.5 | 0.5 |
| 7 | 0.33 | 0.33 | 0.33 |
| 8 | 0.33 | 0.33 | 0.33 |
| 9 | 0.33 | 0.33 | 0.33 |
| 10 | 0.66 | 0.17 | 0.17 |
| 11 | 0.17 | 0.66 | 0.17 |
| 12 | 0.17 | 0.17 | 0.66 |

**Figure 1.** Illustration representing the augmented simplex-centroid design for three components.

As shown by the equation below, the specific cubic model was employed to express the interaction between the components of the analyzed mixtures as well as to predict the maximal recovery of bioactive chemicals and antioxidant activity:

$$Y = \alpha_1 X_1 + \alpha_2 X_2 + \alpha_3 X_3 + \alpha_{12} X_1 X_2 + \alpha_{13} X_1 X_3 + \alpha_{23} X_2 X_3 + \alpha_{123} X_1 X_2 X_3 + \varepsilon \quad (2)$$

Y represents the experimental response expressed by mg GAE/g, mg QE/g and mg/mL for TPC, TFC and DPPH$_{IC50}$, respectively. $\alpha_1$, $\alpha_2$, $\alpha_3$ are linear regression coefficients, $\alpha_{12}$, $\alpha_{23}$ and $\alpha_{23}$ are binary regression coefficients, $\alpha_{123}$: is the ternary regression coefficient, while $\varepsilon$: is the regression error term. The findings were expressed as mean, and ANOVA was used to analyze the results.

### 2.8. Statistical Analysis

Optimization tools were used as an excellent appliance to predict the appropriate combination of extractor solvents. In this context, simplex centroid design was used to assess different interactions among extractor solvents on TPC, TFC and DPPH$_{IC50}$.

The validation of the postulated models was carried out using the F-test for ANOVA. The $F_{R/r}$ (the ratio between the mean square due to the regression ($MS_R$) and the mean square due to the residuals ($MS_r$)) was calculated and then compared to the theoretical F for the same degree of freedom. The lack of fit test was used to refine the model fit by comparing the ratio between the mean square lack of fit ($MS_{LOF}$) and the mean square pure error ($MS_{PE}$) [37]. High values of $F_{LOF/PE}$ indicate a lack of fit to the model. The coefficient of determination $R^2$, $R^2$ adjusted and $R^2$ predicted were used to verify the quality of the postulated models, while Student's test was used to express the significance of the model's coefficients. The experimental design conception and the statistical and graphical analysis

were performed using JMP software V.14 (SAS Institute Inc., Cary, NC, USA) and Design Expert V.11 (Stat-Ease, inc., Minneapolis, MN, USA).

The optimum solvent formula was determined using contour and surface plots based on iso-response curves, which resulted in a compromise of responses. The intensity of the red color indicates the response elevation, while the blue color indicates the opposite. In addition, the desirability function was carried out to accurately determine the desired response based on the optimum conditions. A number of 0% is given when the system produces an undesirable response, while a score of 100% represents the highest possible desired response [38]. Finally, the comparison of means was performed using the ANOVA F-test, while Pearson's test was used to evaluate the correlation between the antioxidant capacity and TPC and TFC. All experiments were performed in triplicate and the results were expressed as the mean $\pm$ SD.

## 3. Results and Discussion

### 3.1. Solvents Screening

Ten solvents were tested to select the most appropriate in maximizing the extractable efficacy on TPC. The results obtained when quantifying phenolic content in terms of used solvents are presented in Figure 2.

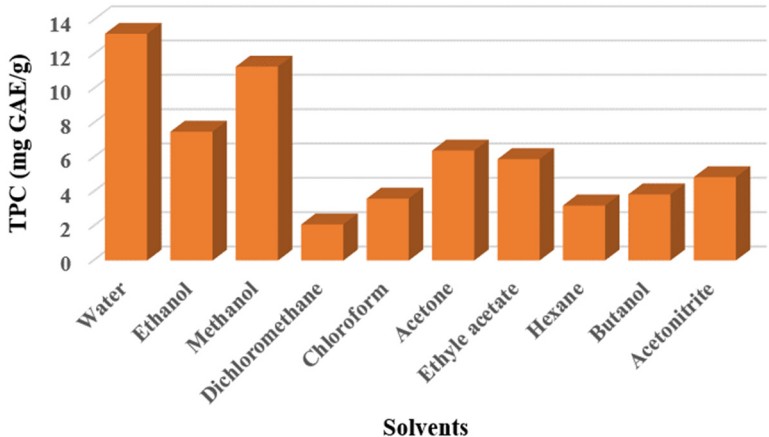

**Figure 2.** Total phenolic compounds extracted from *P. anisum* seeds using different solvents.

The observation of the graph revealed that water was the most suitable solvent to extract the highest quantity of phenolic content with $13.2 \pm 0.36$, followed by methanol and ethanol with $11.29 \pm 0.11$ and $7.5 \pm 0.16$ mg GAE/g, respectively. Acetone and ethyl acetate gave moderate values of TPC with $6.4 \pm 0.62$ and $5.9 \pm 0.79$, respectively. In contrast, dichloromethane extract showed the lowest amount of TPC with only $2.1 \pm 0.1$ mg GAE/g. In addition, the results obtained for the three solvents water, ethanol and methanol were statistically different from the other seven solvents ($p$-value $< 0.05$). The obtained findings were in good concordance with those evoked previously by several researchers for the genius *Rosa canina* L, indicating that high amounts of TPC and TFC were obtained in the aqueous extract compared to ethanolic and methanolic extracts [33,39]. In the same way, Cavalcanti et al. have found that water was the most extractable solvent of TPC with $3.82 \pm 0.07$ mg GAE/g against $0.84 \pm 0.01$ mg GAE/g and $0.35 \pm 0.02$ mg GAE/g obtained by ethanol and acetone, respectively [19]. However, our results did not agree with those provided by Mohsen et al., which indicated that ethanol was the most effective solvent for TPC recovering from corn tassel, followed by methanol and then water [40].

### 3.2. Solvent Mixture

The simplex-centroid design, with different combinations of the solvents and the recorded response of each experiment on TPC, TFC and DPPH$_{IC50}$, are listed in Table 2.

The experiments were conducted after randomization and every response averages three replicates.

**Table 2.** Experimental conditions of mixture design, actual and predicted values and residuals for each response.

| Experiment Number [a] | Solvent's Proportions | | | TPC (mg GAE/g) | | | TFC mg QE/g | | | DPPH$_{IC50}$ (mg/mL) | | |
|---|---|---|---|---|---|---|---|---|---|---|---|---|
| | Water | Ethanol | Methanol | Actual [b] | Predicted | Residual | Actual [b] | Predicted | Residual | Actual [b] | Predicted | Residual |
| 1 | 1 | 0 | 0 | 14.14 ± 0.23 | 13.9 | 0.24 | 4.12 ± 0.72 | 4.12 | 0 | 0.69 ± 0.08 | 0.69 | 0 |
| 2 | 0 | 1 | 0 | 5.62 ± 0.56 | 5.66 | −0.04 | 2.05 ± 0.26 | 2.03 | 0.02 | 1.46 ± 0.06 | 1.47 | −0.01 |
| 3 | 0 | 0 | 1 | 11.29 ± 0.11 | 11 | 0.29 | 3.54 ± 0.41 | 3.51 | 0.03 | 0.77 ± 0.13 | 0.77 | 0 |
| 4 | 0.5 | 0.5 | 0 | 11.84 ± 0.62 | 11.64 | 0.2 | 3.84 ± 0.93 | 3.82 | 0.02 | 0.72 ± 0.05 | 0.74 | −0.02 |
| 5 | 0.5 | 0 | 0.5 | 16.52 ± 0.17 | 15.99 | 0.53 | 4.92 ± 0.12 | 4.89 | 0.03 | 0.68 ± 0.05 | 0.68 | 0 |
| 6 | 0 | 0.5 | 0.5 | 10.6 ± 0.35 | 10.35 | 0.25 | 2.95 ± 0. 38 | 2.9 | 0.05 | 0.97 ± 0.18 | 0.99 | −0.02 |
| 7 | 0.33 | 0.33 | 0.33 | 18.45 ± 0.28 | 17.99 | 0.46 | 7.15 ± 0.56 | 7.17 | −0.02 | 0.6 ± 0.06 | 0.61 | −0.01 |
| 8 | 0.33 | 0.33 | 0.33 | 18.15 ± 0.09 | 17.99 | 0.16 | 7.38 ± 0.67 | 7.17 | 0.21 | 0.59 ± 0.03 | 0.61 | −0.02 |
| 9 | 0.33 | 0.33 | 0.33 | 18.86 ± 0.85 | 17.99 | 0.87 | 7.13 ± 0.18 | 7.17 | −0.04 | 0.57 ± 0.21 | 0.61 | −0.04 |
| 10 | 0.66 | 0.17 | 0.17 | 15.72 ± 0.43 | 16.92 | −1.2 | 5.99 ± 0.87 | 6.04 | −0.05 | 0.6 ± 0.07 | 0.58 | 0.02 |
| 11 | 0.17 | 0.66 | 0.17 | 11.91 ± 0.57 | 12.29 | −0.38 | 4.56 ± 0.43 | 4.67 | −0.11 | 1 ± 0.19 | 0.94 | 0.06 |
| 12 | 0.17 | 0.1 | 0.6 | 14.16 ± 0.92 | 15.53 | −1.37 | 5.39 ± 0.25 | 5.53 | −0.14 | 0.71 ± 0.05 | 0.69 | 0.02 |

[a] Experiments were carried out after randomization. [b] Each response is the average of three replicates with standard error.

The analysis of results revealed that the mixture with equivalent proportions of the three solvents was the most extractable formulation and possessed the highest TPC, TFC and DPPH$_{IC50}$ values (Table 2). Furthermore, the binary mixture of both extractor solvents (water and methanol) with equal volumes showed important TPC values, while pure ethanol was the weakest extractor solvent for TPC yielded, as previously mentioned in the screening of pure solvent extractor efficacy. In fact, no solvent can extract all kinds of bioactive compounds because of their variable solubility and polarity [41].

The efficacy of a mixture of three solvents, water, ethanol and methanol, in extracting the highest content of TPC and TFC is demonstrated by the highest antioxidant ability of the optimized extracts. Based on our findings, the mixture of three solvents was paramount in elevating bioactive components recovery due to the complex composition of plant matrix containing molecules with different polarities [33,42].

This antioxidant activity could be explained by the presence of polyphenols and flavonoids, which have strong antioxidant power in plants [30]. In a study by Bekara et al., the results showed that the aqueous extract of *P. anisum* possesses an important antiradical power [43].

### 3.2.1. Statistical Validation of Postulated Models

The analysis of variance was carried out to study the interaction of different constituents of the mixture, as displayed in Table 3. The results show that the regression's main effect was significant for all studied responses with low *p*-values (0.0011, <0.0001 and <0.0001 for TPC, TFC and DPPH$_{IC50}$, respectively). In addition, the three postulated models do not show a lack of fit since their *p*-values were greater than 0.05 and their calculated $F_{LOF/PE}$ was lower than the theoretical $F_{(0.05;\ 3;2)}$ equal to 19.16 at 95% of confidence.

**Table 3.** The analysis of variance (ANOVA) of studied fitted models.

| Model | DF | TPC | | | | TFC | | | | DPPH$_{IC50}$ | | | |
|---|---|---|---|---|---|---|---|---|---|---|---|---|---|
| | | SS | MS | F | *p*-Value | SS | MS | F | *p*-Value | SS | MS | F | *p*-Value |
| R | 6 | 163.146 | 27.191 | 27.36 | 0.0011* | 33.15 | 5.525 | 326.28 | <0.0001 * | 0.711 | 0.119 | 81.23 | <0.0001 * |
| r | 5 | 4.970 | 0.994 | | | 0.08 | 0.016 | | | 0.007 | 0.001 | | |
| Lof | 3 | 4.71 | 1.57 | 12.37 | 0.0700 | 0.05 | 0.015 | 0.80 | 0.5987 | 0.0068 | 0.0023 | 9.76 | 0.09 |
| Pe | 2 | 0.25 | 0.12 | | | 0.04 | 0.019 | | | 0.0005 | 0.0002 | | |
| total | 11 | 168.11 | | | | 33.23 | | | | 0.72 | | | |
| R$^2$ | | | 0.97 | | | | 0.99 | | | | 0.98 | | |
| R$^2_{adj}$ | | | 0.93 | | | | 0.99 | | | | 0.97 | | |
| R$^2_{pred}$ | | | 0.89 | | | | 0.97 | | | | 0.81 | | |

DF: degrees of freedom; SS: sum of squares; MS: mean square; R: regression; r: residual; Lof: lack of fit; Pe: pure error; adj: adjusted; pred: predicted; *: statistically significant.

The quality of the studied model was confirmed by the coefficient of determination ($R^2$). The latter was equal to 0.97, 0.99 and 0.98 for TPC, TFC and $DPPH_{IC50}$, respectively. The $R^2$ adjusted and predicted are also sufficient to testify the prediction quality of the chosen models. Notably, an excellent correlation was observed between the experimental and predicted results. The graph in Figure 3 confirms this good agreement by showing a linear curve for the actual values in terms of the predicted ones.

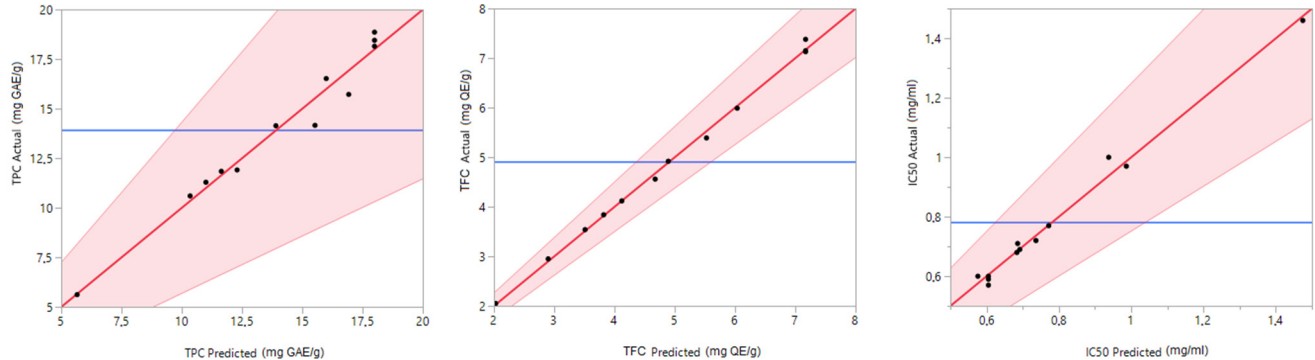

**Figure 3.** Curves of observed values versus predicted ones.

### 3.2.2. Compounds' Effects and Fitted Models

Table 4 summarizes the estimated regression coefficients of the special cubic model. The regression equations with significant coefficients were used to determine the relationships between all studied parameters and the obtained responses for TPC, TFC and $DPPH_{IC50}$).

**Table 4.** Estimated regression coefficients of the special cubic model.

| Term | Coefficient | TPC | | TFC | | $DPPH_{IC50}$ | |
|---|---|---|---|---|---|---|---|
| | | Estimate | *p*-Value | Estimate | *p*-Value | Estimate | *p*-Value |
| Water | $\alpha_1$ | 13.90 | <0.0001 * | 4.12 | <0.0001 * | 0.69 | <0.0001 * |
| Ethanol | $\alpha_2$ | 5.66 | 0.0020 * | 2.03 | <0.0001 * | 1.47 | <0.0001 * |
| Methanol | $\alpha_3$ | 11.00 | <0.0001 * | 3.51 | <0.0001 * | 0.77 | <0.0001 * |
| Water*Ethanol | $\alpha_{12}$ | 7.44 | 0.19 | 2.98 | 0.0053 * | −1.39 | 0.0007 * |
| Water*Methanol | $\alpha_{13}$ | 14.16 | 0.0330 * | 4.30 | 0.0010 * | −0.19 | 0.35 |
| Ethanol*Methanol | $\alpha_{23}$ | 8.07 | 0.16 | 0.52 | 0.45 | −0.55 | 0.0321 * |
| Water*Ethanol*Methanol | $\alpha_{123}$ | 121.82 | 0.0057 * | 83.25 | <0.0001 * | −3.73 | 0.0141 * |

* Statistically significant at *p* < 0.05.

The significant extractor solvent coefficients for TPC are $\alpha_1$, $\alpha_2$ and $\alpha_3$, for pure extractor solvents, followed by the coefficient of the binary formulation between water and methanol ($\alpha_{13}$) and then the ternary term ($\alpha_{123}$). The following equation represents the mathematical model adopted for TPC:

$$Y_{TPC} = 13.90X_1 + 5.66X_2 + 11X_3 + 14.16X_{13} + 121.82X_{123} + \varepsilon \tag{3}$$

Concerning TFC, the significant terms are all the terms of the adapted mathematical model ($\alpha_1$, $\alpha_2$, $\alpha_3$, $\alpha_{12}$, $\alpha_{13}$ and $\alpha_{123}$), except the coefficient that corresponds to the binary combination between ethanol and methanol ($\alpha_{23}$). The following equation is used to represent the retained mathematical model for TFC:

$$Y_{TFC} = 4.12X_1 + 2.03X_2 + 3.51X_3 + 2.98X_{12} + 4.3X_{13} + 83.25X_{123} + \varepsilon \tag{4}$$

Regarding DPPH$_{IC50}$, all terms are significant except the coefficient corresponding to the binary term $\alpha_{13}$. DPPH$_{IC50}$ response was determined using the following mathematical model:

$$Y_{DPPH_{IC50}} = 0.69X_1 + 1.47X_2 + 0.77X_3 - 1.39X_{12} - 0.55X_{23} - 3.73X_{123} + \varepsilon \qquad (5)$$

### 3.2.3. Solvent System Optimization
Total Polyphenolic Content

Figure 4 presents the contour and surface plots realized for total polyphenolic content obtained using a different mixture of water, ethanol and methanol.

The TPC of 12 different extracts of the samples ranged from 5.62 to 18.86 mg GAE/g (Table 2). The illustrations in the contour and surface plots (Figure 4A) show that a mixture of water, methanol and ethanol is needed to reach a value of around 18 mg GAE/g. Furthermore, the desirability plot (Figure 5A) shows that the maximum value of the TPC that can be achieved is equal to 18.5 mg GAE/g. This value is possible with a desirability of 99% by ensuring the following operating conditions (40:23:37 *v/v/v*) water:ethanol:methanol.

The obtained TPC was higher than previous studies that used only simple solvents, as in the case of Wang et al., who obtained only 7.50 ± 0.11 mg GAE/g DW using methanol and 10.20 ± 0.09 mg GAE/g DW using aqueous extracts [44]. The only acceptable reason to explain the higher quantity of polyphenol content in this study is that the combined effects (synergy) of the used solvents counting water, ethanol and methanol are superior to simple methods of dissolving polyphenols. Similar results were revealed in other plants, such as *Coffea arabica* L. and *Spondias mombin* L., in which mixture design recorded superior results for polyphenols as compared to simple methods [45,46]. In another study, Herrera-Pool et al. have found that the highest recovery of TPC from *Capsicum chinense* was obtained using methanol 50% by ultrasound-assisted extraction [47]. With the same target of maximizing the total phenolic compounds recovery, Šaponjac et al. found that the most appropriate combination to maximize the recovery of phenolic compounds from carrot required mixing 20% of water, 49% acetone and 31% ethanol [48]. Besides, a solvent system consisting of (53:35:12 *v/v/v*) methanol:ethanol:acetone ternary mixture was found to be the most adequate to recover a high amount of TPC from Moroccan *Lavendula stoechas* [49].

Total Flavonoid Content

Thanks to Figure 4B, which displays the contour and surface plots of the response TFC, we can see that a content of 7 mg QE/g was registered for the ternary combination with equal proportions of three studied solvents containing water-ethanol-methanol (*v/v/v*). In addition, the desirability function indicates that the precise optimal amount of TFC was equal to 7.25 mg QE/g with a compromise percentage of 99% (Figure 5B).

These results were significantly superior to those of Balanescu et al., who found 5.55 ± 0.42 mg QE/g TFC using the Ultrasound-Assisted Extraction method [50]. The effect of mixture solvent on TFC was previously reported for extra virgin olive oil and *Pimpinella barbata (dc.)* Boiss. *Jundishapur*. [51,52]. Moreover, in the study conducted by Fadil et al., a solvent system containing (40:27:33 *v/v/v*) ethyl acetate:methanol:ethanol was the best-recommended mixture to optimize total flavonoids content from *Lavandula stoechas* [49].

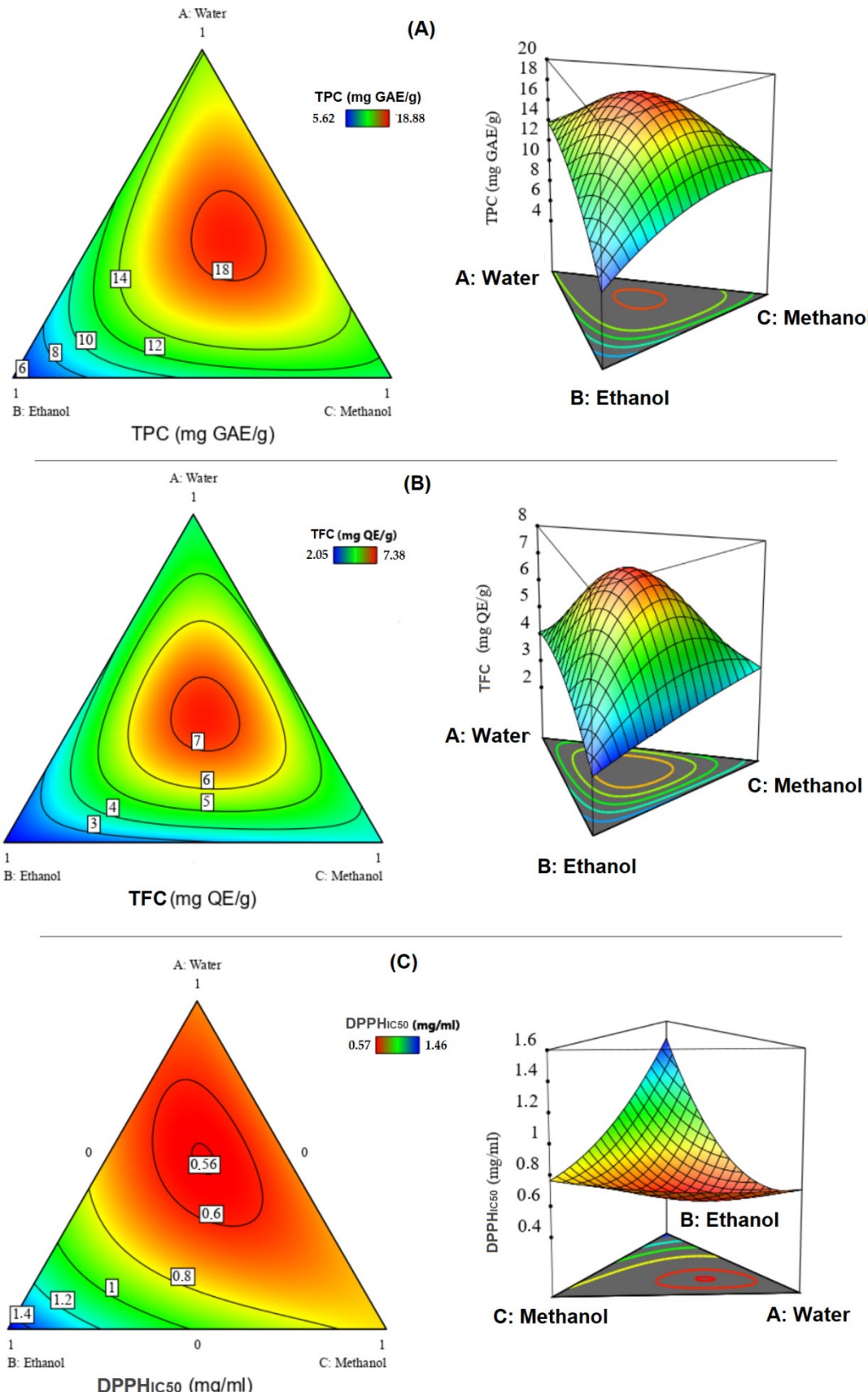

**Figure 4.** Contour and surface plots, which indicate the optimal compromise area leading to the best value of TPC (**A**), TFC (**B**) and DPPH$_{IC50}$ (**C**).

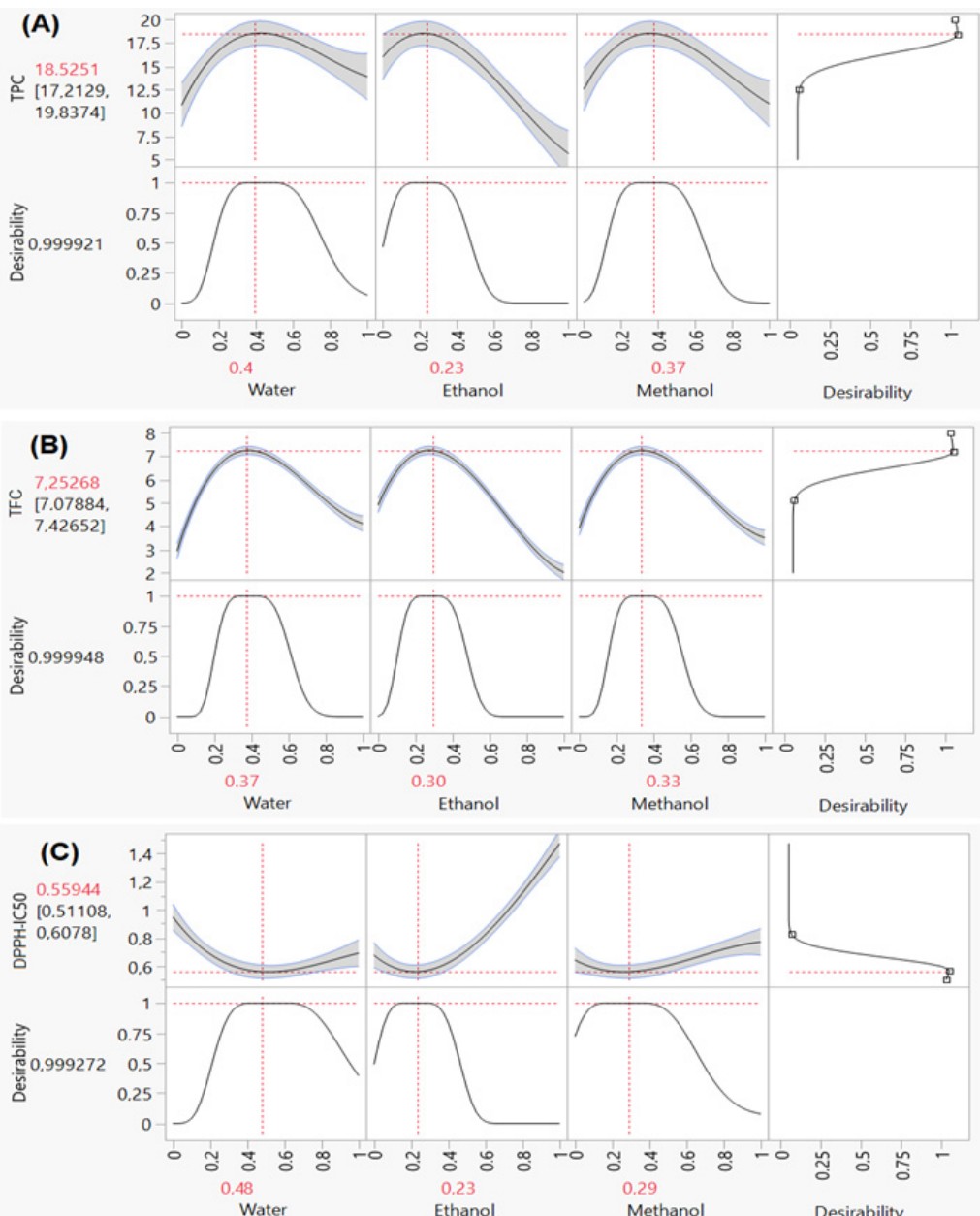

**Figure 5.** Desirability plots used to identify the optimum value of TPC (**A**), TFC (**B**) and DPPH$_{IC50}$ (**C**), separately.

Antioxidant Activity

The efficacy of recovering phytochemicals with high antioxidant potential constitutes an important step to elevate the beneficial properties of the optimized extracts. Within this background, the main objective of the current study was to determine the efficacy of the simplex centroid mixture design in finding the most effective mixture to extract the highest amount of bioactive ingredients with antioxidant potential. The analysis results of the contour and surface plots (Figure 4C) revealed that we could reach a value of DPPH$_{IC50}$ around 0.56 mg/mL by ensuring the ternary combination with water, ethanol and methanol. Moreover, the desirability test in Figure 5C showed that the ternary mixture with the following proportion (48:23:29 *v/v/v*) water:ethanol:methanol gives the best antioxidant ability. In the same way, Aazza et al. have reported that the solvent mixture containing 25% ethanol and 75% methanol was the most appropriate solution to improve the antioxidant ability of *Cannabis sativa* Waste [30], while the binary mixture 75:25 of water:acetone was

better for the DPPH scavenging activity of *Allium sativum L.* [19]. In a recent study, Fadil et al. demonstrated that the ternary mixture (50:36:14 *v/v/v*) methanol:ethanol:acetone was the optimal solvent system for better optimizing the antioxidant power of Moroccan *Lavandula stoechas* [49].

Simultaneous Optimization of all Responses

Depending on the priority of the responses under study, the simultaneous optimization of all responses using the desirability function could provide various solutions. In our case, the response DPPH$_{IC50}$ was the most important. Thus, simultaneous optimization involves finding the best compromise to improve all responses with the priority of the DPPH$_{IC50}$ response. According to the desirability plot (Figure 6), the maximum responses for TPC, TFC and antioxidant activities by DPPH assays were obtained by the ternary solvents mixture (44:22:34 *v/v/v*) water:methanol:ethanol. With this solvent system, we can obtain a value of 18.55 mg GAE/g, 7.16 mg QE/g and 0.56 mg/mL for TPC, TFC and DPPH$_{IC50}$, respectively. The mixture contour plot of all the responses, created by the three solvents water, methanol and ethanol (Figure 7), makes the effects of the simultaneous optimization more obvious. The position of these three responses in the combined mixture plot suggests a possible correlation between them. Indeed, it has been previously demonstrated that a statistical correlation exists between the amount of total phenol and flavonoid content and the antioxidant activity [49]. In this way, a correlation test was performed to check this hypothesis. As a result, and since the plant extracts with higher levels of total phenolics exhibit low values of DPPH$_{IC50}$, a significant negative correlation was observed for the correlation between DPPH$_{IC50}$ with TPC (r = −0.89; *p*-value = 0.0001) and TFC(r = −0.78; *p*-value = 0.028).

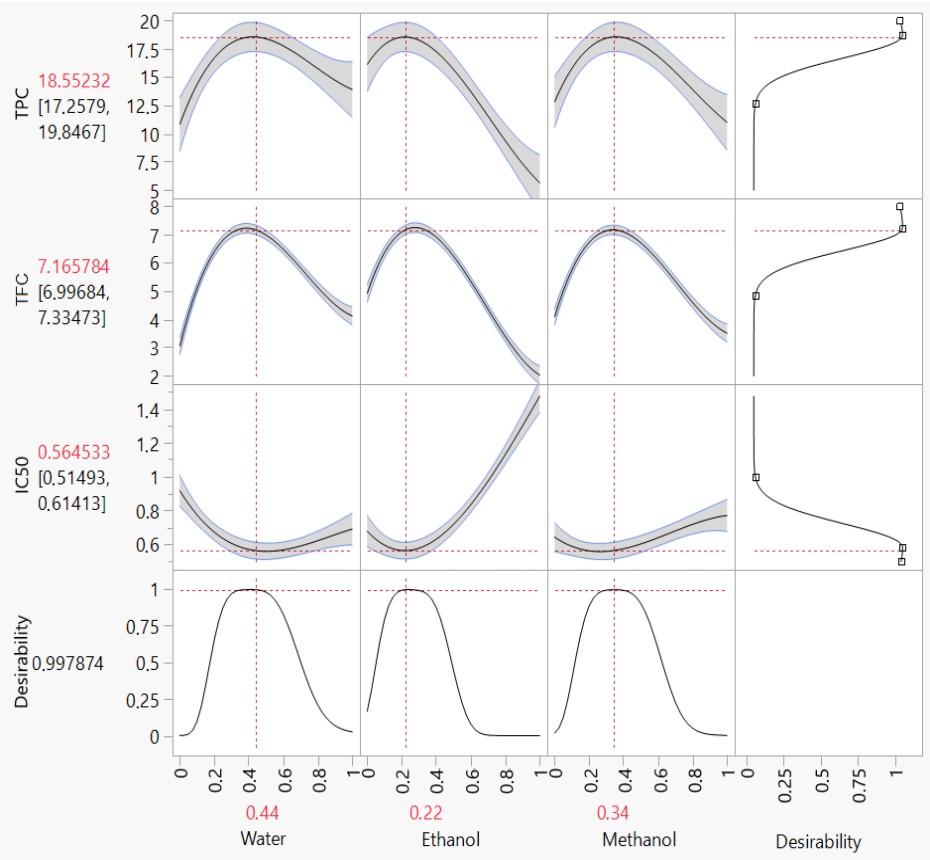

**Figure 6.** Desirability plots of simultaneous optimization demonstrating the maximum value of TPC, TFC and DPPH$_{IC50}$.

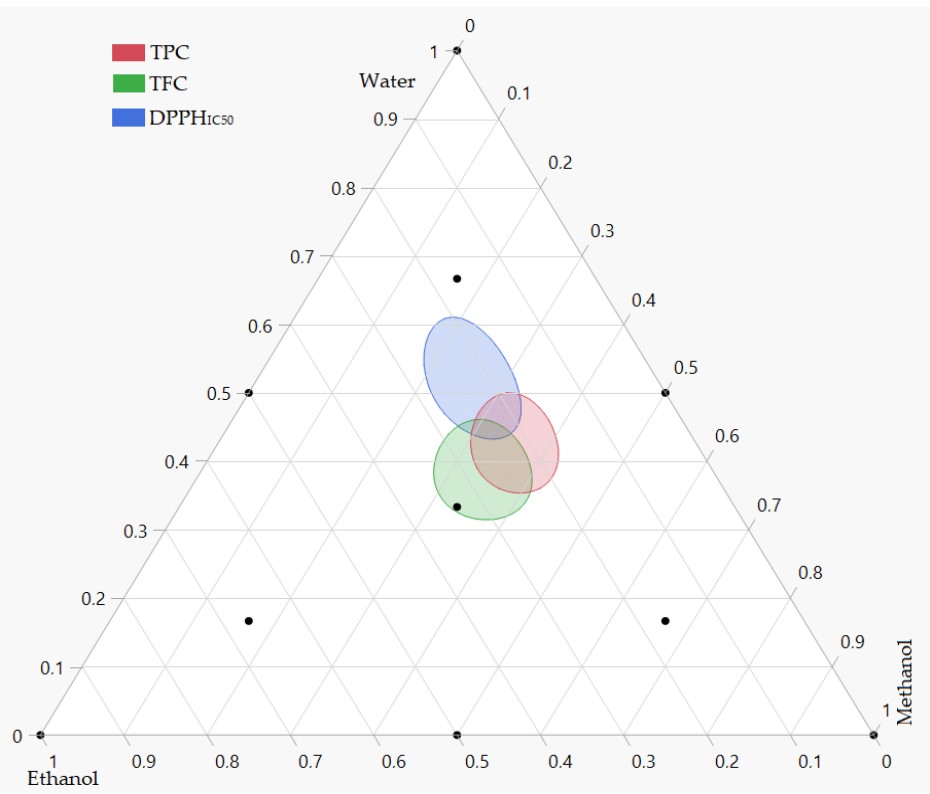

**Figure 7.** Mixture contour plot illustrating the simultaneous optimization of the three responses based on the values given by desirability functions.

Experimental Validation of the Optimal Conditions

The current study's final step was the validation of the generated special cubic models. The suitability of the prediction model was evaluated by conducting an experiment under appropriate conditions and comparing predicted and experimental values. The chosen test point coordinates were the proportions of the optimal solvent system obtained for simultaneous optimization. Table 5 shows that the observed and predicted values are concordant, demonstrating that the postulated models' choice has been judicious.

**Table 5.** Predicted and experimental values for the test point realized by the optimal found mixtures.

| Compounds of Mixture | Proportions of Solvents (%) | TPC mg GAE/g | | TFC mg QE/g | | DPPH$_{IC50}$ (mg/mL) | |
| --- | --- | --- | --- | --- | --- | --- | --- |
| | | Predicted [a] | Experimental [b] | Predicted [a] | Experimental [b] | Predicted [a] | Experimental [b] |
| Water | 44 | | | | | | |
| Ethanol | 22 | $18.55 \pm 1.29$ | $19.01 \pm 0.23$ | $7.16 \pm 0.17$ | $7.02 \pm 0.12$ | $0.56 \pm 0.05$ | $0.56 \pm 0.01$ |
| Methanol | 34 | | | | | | |

[a] The predicted value is given with the standard deviation of the response calculated from the model. [b] The observed value is the average of three replicates with standard error.

## 4. Conclusions

In the current study, the optimization of the extraction of bioactive compounds from anise was performed using a simplex-centroid design. A solvent system containing water, ethanol and methanol was optimized to improve the three responses, TPC, TFC and DPPH$_{IC50}$. The validated special cubic models allowed the optimization of all responses simultaneously. Thus, the findings indicated that the ternary mixture of water:ethanol:methanol (44:22:33 $v/v/v$) was the most appropriate for simultaneous maximization of TPC, TFC and antioxidant activity. The optimized responses were 18.55 mg GAE/g, 7.16 mg QE/g and 0.56 mg/mL for TPC, TFC and DPPH$_{IC50}$, respectively. These

conclusions could be a key development in managing the extraction of phenolic compounds for specific applications, such as their application as a substitute for traditional antioxidants utilized mostly in the food industry to enhance nutrition quality. In addition, our work has demonstrated that mixture designs are a valuable and effective tool for organizing and refining experimental parameters to obtain the best findings with the fewest experiments possible.

**Author Contributions:** Conceptualization, M.S. and M.F.; methodology, M.S. and M.F. Software, validation, formal analysis M.F. Writing—original draft preparation, M.S., W.A.Y. and M.F. Writing—review and editing, M.S., M.F. and M.B.; Visualization, L.E.G., M.F. and M.B.; supervision, M.F. and L.E.G. All authors have read and agreed to the published version of the manuscript.

**Funding:** This research received no external funding.

**Informed Consent Statement:** Not applicable.

**Data Availability Statement:** The data used to support the findings of this study are included within the article.

**Conflicts of Interest:** The authors declare no conflict of interest.

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
