# Peer review of "Simultaneous Optimization of Phenolic Compounds and Antioxidant Abilities of Moroccan Pimpinella anisum Extracts Using Mixture Design Methodology"

_processes, doi:10.3390/pr10122580_

Round 1

Reviewer 1 Report

The manuscript's overall English is very poor. Sentence structure, word selection and grammar have to be substantially improved. Identification of key phenolics in the optimised extract in comparison to the untreated extract using HPLC-UV-Vis or LC-MS is required. 

Author Response

Reviewer #1

Comment 1

The manuscript's overall English is very poor. Sentence structure, word selection and grammar have to be substantially improved.

Reply:

We thank the reviewer for this query. As suggested, we have tried our best to improve the level of English in the whole article.

Comment 2

Identification of key phenolics in the optimised extract in comparison to the untreated extract using HPLC-UV-Vis or LC-MS is required.

Reply:

We thank the reviewer for this pertinent remark, which will undoubtedly improve the quality of our manuscript. However, technical constraints in accessing the apparatuses such as HPLC-DAD or LC-MS prevented us from finalizing this identification step.

Reviewer 2 Report

The manuscript titled: Simultaneous optimization of phenolic compounds and antioxidant abilities of Moroccan Pimpinella anisum extracts using 4 mixture design methodology was reviewed. The manuscript has serious problems on the methodology, grammar, and  references, which requires extensive revision. In addition, the authors should make an effort to adequately discuss their results with reference to literature. Below are additional comments.

Title: The expression Pimpinella anisum should be written in italics

Abstract: DPPH, TFC, and TPC should be written in full before shortening, particularly when being mentioned for the first time in the manuscript.

Introduction: the introduction needs to be improved to facilitate reading. For instance, the criteria of selecting the 10 solvents used was not explained in the introduction

Materials and methods

 Line 80. Material et methods should be revised to the correct expression

Line 81-83. The information should be removed it does not add value to the manuscript

Line 84. The authors should add the source of anise seeds under chemicals and materials section

Line 87. The Sigma Aldrich was from which country? Add the information.

Line 89-91. I suggest that this information be moved to chemicals and materials section

Line 92. The authors should add the temperature range of the room in which the anise seeds where dried

Line 96-97. The authors should add the model, city, province, and country from which the Sonicator and Centrifuge were manufactured.

Line 105. ……………………and as the mean ± SD from…. The authors should revise this  statement to make it clearer

Line 108. …………………The dosage of total flavonoid content…… The authors should revise this  statement to make it clearer

Line 121, equation 1. The authors should describe what PI represent

Line 126. The mixture is not clear, for instance, the authors should add a table showing the different proportions of the solvents used. The 3 factors used in the design where not clearly stated.

Line 147. Expert Design software v.11? I don’t think this is the correct name for the software. The authors should also add the city and countries of the software used.

Line 149. Besides the R2 did the authors get R2 adjusted, R2 predicted, and lack of fit tests results from the output. These are important to determine the fitness of the models and should be clear in the methodology

Line 151-158. How were the models validated? This should be clear in the methodology

Line 160. The way the solvents were screened is not included in the methodology

Results and discussion

Line 160. The screening results are not clearly presented or reported. The authors

Line 177. Figure 1 is poorly presented.

Line 184. The authors should replace commas in Table 1 with dots or full stops.

Line 225. What do the authors mean by water (mixture), ethanol (mixture)…………in table 3?

Line 248: The figure caption should be revised. E.g., it states contour plot only and leave out surface plots, there is no need to keep repeating the solvents on each measurement

Line 342. What do the superscript letters a and b in Table 4 mean?

Conclusion

The conclusion is poorly written. It lacks the most important findings and reference to the broader aim of the study. The authors should try to link their findings to Anise seeds and how their findings are important in solving the broader aim of the study

Author Response

Reviewer #2

Comment 1

Title: The expression Pimpinella anisum should be written in italics.

Reply:

We thank the reviewer for this suggestion; as recommended, all scientific names were given in italics.

Comment 2

DPPH, TFC, and TPC should be written in full before shortening, particularly when being mentioned for the first time in the manuscript..

Reply:

The reviewer's suggestion was introduced in the manuscript.

Comment 3

The introduction needs to be improved to facilitate reading. For instance, the criteria of selecting the 10 solvents used was not explained in the introduction.

Reply:

The criteria and purpose of the screening step were added to the introduction as suggested by the reviewer.

Comment 4

Line 80. Material et methods should be revised to the correct expression

Reply:

The suggested correction was made.

Comment 5

Line 81-83. The information should be removed it does not add value to the manuscript.

Reply:

The suggested correction was made.

Comment 6

Line 84. The authors should add the source of anise seeds under chemicals and materials section.

Reply:

The suggested correction was made.

Comment 7

Line 87. The Sigma Aldrich was from which country? Add the information.

Reply:

The suggested information was added.

Comment 8

Line 89-91. I suggest that this information be moved to chemicals and materials section.

Reply:

The suggested correction was made.

Comment 9

Line 92. The authors should add the temperature range of the room in which the anise seeds where dried

Reply:

The drying of the anise seeds was carried out in a chamber in which the temperature was fixed at 25 °C. This information was added in the text.

Comment 10

Line 96-97. The authors should add the model, city, province, and country from which the Sonicator and Centrifuge were manufactured.

Reply:

The requested information about Sonicator and Centrifuge was introduced in the manuscript

Comment 11

Line 105. ……………………and as the mean ± SD from…. The authors should revise this  statement to make it clearer

Reply:

The concerned statement was corrected and added in the statistical analysis section.

Comment 12

Line 108. …………………The dosage of total flavonoid content…… The authors should revise this  statement to make it clearer

Reply:

This statement was modified to make it clearer.

Comment 13

Line 121, equation 1. The authors should describe what PI represent

Reply:

PI means the percentage of inhibition. This abbreviation was added in the text.

Comment 14

Line 126. The mixture is not clear, for instance, the authors should add a table showing the different proportions of the solvents used. The 3 factors used in the design where not clearly stated.

Reply:

As suggested by the reviewer, a table summarizing all the constituents of the solvent system and a figure representing the augmented simplex-centroid design experiments were added in the mixture design section

Comment 15

Line 147. Expert Design software v.11? I don’t think this is the correct name for the software. The authors should also add the city and countries of the software used.

Reply:

The name of the software was corrected, and the statement that mentions the details of the software used was reformulated to make it clearer.

Comment 16

Line 149. Besides the R2 did the authors get R2 adjusted, R2 predicted, and lack of fit tests results from the output. These are important to determine the fitness of the models and should be clear in the methodology

Reply:

The reviewer's suggestions were introduced in the section statistical analysis

Comment 17

Line 151-158. How were the models validated? This should be clear in the methodology

Reply:

The steps involved in validating the postulated models are described in more detail in the statistical analysis section.

Comment 19

Line 160. The way the solvents were screened is not included in the methodology

Reply:

The way the solvents were screened was inserted in paragraph 2.3. Ultrasonic-assisted extraction

Comment 20

Line 160. The screening results are not clearly presented or reported.

Reply:

As the reviewer suggested, the screening results' presentation was redone and their discussion was improved.

Comment 21

Line 177. Figure 1 is poorly presented

Reply:

Figure 1 was redone for better presentation

Comment 22

Line 184. The authors should replace commas in Table 1 with dots or full stops.

Reply:

The suggested correction was made in all the tables.

Comment 23

Line 225. What do the authors mean by water (mixture), ethanol (mixture)…………in table 3?

Reply:

This is a manner in which the software presents the tables, the word (mixture) was removed from the table.

Comment 24

Line 248: The figure caption should be revised. E.g., it states contour plot only and leave out surface plots, there is no need to keep repeating the solvents on each measurement.

Reply:

The suggested correction was made.

Comment 25

The conclusion is poorly written. It lacks the most important findings and reference to the broader aim of the study. The authors should try to link their findings to Anise seeds and how their findings are important in solving the broader aim of the study

Reply:

The conclusion was redone according to the reviewer's suggestions

Reviewer 3 Report

1.      There is a need in the thorough formatting of the manuscript (spelling, font, usage of scientific names, etc).

2.      Line # 98, 4 C° should be 4 °C

3.      Line # 121, in the formula, Controle should be control.

4.      Line #184, Table 1, 14,14±0.23 should be14.14±0.23 (instead of comma it should be decimal point). Check for the similar mistakes throughout the manuscript (even in the Tables 2, 3 and 4).

5.      Line # 301 Cannabis Sativa Waste should be Cannabis sativa waste. Species name should be in small letters. Check out the similar mistakes.

6.      Line # 482, Reference # 49 in capital letters; check it out throughout the reference section for the same mistake.

7.      Mainly authors should concentrate on the particular phenolic compounds of the plant such as naringin, gallic, rosmarinic, ellargic, syringic acids etc. instead of crude extract. It would throw some scientific interest. I would suggest authors have to take minimum two phenolic compounds and optimize the content.

Author Response

Reviewer #3

Comment 1

There is a need in the thorough formatting of the manuscript (spelling, font, usage of scientific names, etc).

Reply:

We thank the reviewer for this remark, as suggested, the article's formatting was revised.

Comment 2

Line # 98, 4 C° should be 4 °C

Reply:

As suggested, the correction was made.

Comment 3

Line # 121, in the formula, Controle should be control.

Reply:

The correction was made as suggested by the reviewer.

Comment 4

Line #184, Table 1, 14,14±0.23 should be14.14±0.23 (instead of comma it should be decimal point). Check for the similar mistakes throughout the manuscript (even in the Tables 2, 3 and 4).

Reply:

Commas were changed to points in all tables of the manuscript

Comment 5

Line # 301 Cannabis Sativa Waste should be Cannabis sativa waste. Species name should be in small letters. Check out the similar mistakes.

Reply:

The correction was introduced in the text and all species names were checked

Comment 6

Line # 482, Reference # 49 in capital letters; check it out throughout the reference section for the same mistake.

Reply:

The correction was introduced in the references as suggested by the reviewer

Comment 7

Mainly authors should concentrate on the particular phenolic compounds of the plant such as naringin, gallic, rosmarinic, ellargic, syringic acids etc. instead of crude extract. It would throw some scientific interest. I would suggest authors have to take minimum two phenolic compounds and optimize the content.

Reply:

We thank the reviewer for this relevant comment, which could have enhanced the work's quality. However, as we have already explained, we had technical constraints in accessing characterization equipment such as HPLC-DAD and LC-MS. This prevented us from going further in the characterization of our extracts and/or their optimization.

Reviewer 4 Report

My all observations have been pointed out in the manuscript itself. Needs proper editing of English and uniformity in the scientific name. Improve all figures having more and clear visibility to the readers. You may enhance their quality and size.

Author Response

Reviewer #4

Comment 1

Italic

Reply:

All scientific names of species were checked and italicized

Comment 2

Delete, not required

Reply:

The concerned section was deleted

Comment 3

Discussion part is insufficient. Discuss your results according to the references 

Reply:

The discussion of findings was improved as suggested by the reviewer.

Comment 4

no detailed information about these

Reply:

Detailed information on solvent screening was added in the materials and methods section

Comment 5

Use uniform pattern in whole manuscript. You used L here but in title and introduction.

Here you can use only P as genus name.

Reply:

The reviewer's suggestion was taken into consideration, a uniform pattern was used in the whole manuscript.

Comment 6

Make figures more clear and enlarge so that readers understand properly.

Reply:

The quality of the figures was good in their separate state; however, the insertion of the figures in the text decreased their quality. Therefore, we have tried to improve the quality of all the figures to be more readable.

Comment 7

Conclusion is not proper

Reply:

The conclusion was redone according to the reviewer's suggestions

Round 2

Reviewer 2 Report

1. The introduction needs to be improved to facilitate reading. For instance, the criteria for selecting the 10 solvents used were not explained in the introduction. The question is why did the authors decide to use these solvents to extract the phenolic compounds among all the available solvents, and did the authors consider the detrimental effects of some of the solvents such as methanol, dichloromethane, acetone, and hexane on human health considering that the potential application of their product is in phamarceuticals?  

2. What do levels 0 and 1 in table 1 represent? they should be described.

3. Did the authors perform statistical analyses on the TPC results in Figure 2 to determine if the TPC content were significantly different before making a decision to select the 3 solvents? 

4. The references still need to be corrected throughout the manuscript

Author Response

Reviewer #2

The introduction needs to be improved to facilitate reading. For instance, the criteria for selecting the 10 solvents used were not explained in the introduction. The question is why did the authors decide to use these solvents to extract the phenolic compounds among all the available solvents, and did the authors consider the detrimental effects of some of the solvents such as methanol, dichloromethane, acetone, and hexane on human health considering that the potential application of their product is in phamarceuticals? 

Reply:

We thank the reviewer for this pertinent suggestion. In fact, the solvents chosen are all used to extract phenolic compounds from plant matrices. Of course, some properties influence the extractive effect from one solvent to another, but the key criterion is the polarity. Thus, the solvents were chosen to have a gradient of polarity from the least polar to the most polar. Thus, we started from apolar solvents such as hexane to very polar solvents such as water and methanol. This clarification was introduced in the introduction to justify the choice of solvents.

Comment 2

What do levels 0 and 1 in table 1 represent? they should be described.

Reply:

0 and 1 are the coded variables of 0% and 100%. As added in the manuscript, each of the three components of the mixture can have a value between 0 and 100%, and the sum of the three proportions will always be equal to 100%. The real values of constituents were added to table 1 to make it clearer.

Comment 3

Did the authors perform statistical analyses on the TPC results in Figure 2 to determine if the TPC content were significantly different before making a decision to select the 3 solvents?

Reply:

We thank the reviewer for this pertinent remark. Indeed, since the tests were performed in three repetitions, it allowed us to perform a comparison of means by F-ANOVA test and the differences were statistically significant between the values observed for the three selected solvents and the other solvents. As recommended, the results of this statistical test have been introduced in the paragraph "Solvents screening". In addition, the used statistical test was added to the statistical analysis section.

Comment 4

The references still need to be corrected throughout the manuscript.

Reply:

All references have been checked and corrected.

Reviewer 3 Report

1.   Line # 150, Figure 1, Authors are using commas, instead of decimal point. Similar mistakes were also observed in Fig. 3 to 7.

2. I strongly suggest the authors have to carry out the experiment on minimum two phenolic compounds and optimize the content.

Author Response

Reviewer #3

Comment 1

Line # 150, Figure 1, Authors are using commas, instead of decimal point. Similar mistakes were also observed in Fig. 3 to 7..

Reply:

We thank the reviewer for this remark, as suggested, the correction was made.

Comment 2

I strongly suggest the authors have to carry out the experiment on minimum two phenolic compounds and optimize the content.

Reply:

We thank the reviewer again for his pertinent suggestions. Unfortunately, and as already explained. We currently have a lot of constraints in accessing an instrument for identification and quantification of phenolic compounds but we will keep this important remark in our ongoing work.

Reviewer 4 Report

This is fine for me. 

Author Response

Thank you very much for your review and feedback.